# Surface Functionalization of Black Phosphorus via Amine Compounds and Its Impacts on the Flame Retardancy and Thermal Decomposition Behaviors of Epoxy Resin

**DOI:** 10.3390/polym13213635

**Published:** 2021-10-21

**Authors:** Shaoling Lin, Boqing Tao, Xiaomin Zhao, Guohua Chen, De-Yi Wang

**Affiliations:** 1College of Chemical Engineering, Huaqiao University, Xiamen 361021, China; 15960378351@163.com; 2College of Materials Science and Engineering, Huaqiao University, Xiamen 361021, China; luoyileng@163.com (B.T.); hdcgh@hqu.edu.cn (G.C.); 3IMDEA Materials Institute, C/Eric Kandel, 2, 28906 Getafe, Spain; 4Escuela Politécnica Superior, Universidad Francisco de Vitoria, Ctra. Pozuelo-Majadahonda Km 1,800, 28223 Pozuelo de Alarcón, Spain

**Keywords:** black phosphorus, functionalization, flame retardancy, epoxy resin

## Abstract

Recently, lots of effort has been placed into stabilizing black phosphorus (BP) in the air to improve its compatibility with polymers. Herein, BP was chemically functionalized by aliphatic amine (DETA), aromatic amine (PPDA) and cyclamine (Pid) via a nucleophilic substitution reaction, aiming to develop an intensively reactive BP flame retardant for epoxy resin (EP). The -NH_2_ group on BP-DETA, BP-PPDA and BP-Pid reacted with the epoxide group at different temperatures. The lowest temperature was about 150 °C for BP-DETA. The impacts of three BP-NH_2_ were compared on the flame retardancy and thermal decomposition of EP. At 5 wt% loading, EP/BP-NH_2_ all passed UL 94 V 0 rating. The limiting oxygen index (LOI) of EP/BP-PPDA was as high as 32.3%. The heat release rate (HRR) of EP/BP-DETA greatly decreased by 46% and char residue increased by 73.8%, whereas HRR of EP/BP-Pid decreased by 11.5% and char residue increased by 50.8%, compared with EP. Average effective heat of combustion (av-EHC) of EP/BP-Pid was lower than that of EP/BP-DETA and EP/BP-PPDA. In view of the flame-retardant mechanism, BP nanosheets functionalized with aliphatic amine and aromatic amine played a dominant role in the condensed phase, while BP functionalized with cyclamine was more effective in the gas phase.

## 1. Introduction

Epoxy resin (EP), an important thermosetting polymeric material, is widely used in many fields, such as in coating, construction, transportation, electronic and electrical industries (EE) due to its excellent electrical insulation performance, high mechanical strength and good chemical resistance [1,2,3,4,5]. In general, EE products require a flame-retardancy grade to guarantee its safety of use in heated environment. With a limiting oxygen index (LOI) of only 20%, flammability is an obvious drawback for common EP, e.g., bisphenol A-EP, which is widely used at present. Therefore, flame-retardant modification for EP is significant for human life and property. Phosphorus-containing flame retardants are becoming one of the most important research branches in the field of flame retardants, owing to its advantages of a high efficiency, halogen-free and eco-friendly [6,7,8,9].

As a two-dimensional (2D) nanomaterial, black phosphorus (BP) has found many potential applications in the fields of electronic, photonic, medical and energy storage devices, owing to its distinct physical and chemical properties [10]. BP, the most thermodynamically stable allotrope of phosphorous, has an orthogonal puckered layer [11], which is formed of P atoms covalently bonded to another three P atoms. Meanwhile, BP has a large specific surface area, a peelable layered structure, and is composed of a flame retardant element P, making it gradually developed in the field of halogen-free flame retardancy [12,13,14,15,16,17,18]. Although BP is excellent in many ways, chemical degradation of phosphorous into phosphorus oxides in the presence of ambient oxygen and water results in the rapid loss of properties, due to the high reactivity of the lone pair of electrons in BP [19,20,21,22,23,24,25,26,27,28,29]. Moreover, BP has poor compatibility with the polymer material. The method of direct addition of BP into resins would cause serious agglomeration, which greatly reduces the flame retardant efficiency of BP.

To improve the environmental stability of BP, a number of essential strategies, including protective layer coating [30,31,32,33,34,35,36,37,38,39] and chemical modification [40,41,42,43,44,45,46,47], have been employed to suppress the chemical degradation of exfoliated BP. For example, Al_2_O_3_ protection can preserve the properties of BP over 7 days of ambient exposure; however, long-term stability is still challenging [33,37]. Chemical functionalization utilizes the lone pair of electrons present in the phosphorous atom to form direct chemical bonds and can thus protect BP from oxygen, to achieve long-term stability for BP at ambient conditions [22]. The covalent functionalization methods, involving radical addition, nitrene addition, nucleophilic substitution and metal coordination, have been demonstrated to be the most effective. On the one hand, the environmental stability of BP is improved by this method. On the other hand, if the amino reactive groups are grafted, the dispersion of BP in epoxy resin can be improved, so as to solve the aggregation problem of BP. Numerous researchers have done many studies on the surface of amino functional modification of BP and have made some progress [17,44]. However, little attention has been paid to the reactivity of surface amino groups in the existing literature, which is a crucial prerequisite to realize the above two modification purposes.

In this work, three typical amine-contain compounds (aliphatic amine, aromatic amine and cyclamine) were chosen to functionalize BP via the nucleophilic substitution reaction (Figure 1). The reactivity of amino groups of BP-NH_2_ and epoxy groups of epoxy resin were studied respectively. The influence of the modified structure on the thermal stability of BP was also studied. The relationship between the amino structures and the flame retardancy efficiency of condensed-gas phases of BP in the EP matrix was compared. The results showed that, compared with aromatic amine and cyclamine, the amino group of BP modified by aliphatic amine had the highest reactivity with the epoxy group and the reaction temperature was about 150 °C. From the results of the combustion tests, EP/BP-NH_2_ nanocomposites could pass the UL 94 V 0 rating with the same amount of 5 wt% flame retardant. The limiting oxygen index (LOI) value of EP/BP-PPDA (5 wt%) was as high as 32.3%, the residual char rate increased by 70.6%. Moreover, the heat release rate (HRR) and the total heat release (THR) decreased by 39.8% and 51.7%, which was assigned to the restriction of heat transfer and inhibition of flammable gas by the dense char residues. Among the EP/BP-NH_2_ (5 wt%) nanocomposites, the EP/BP-Pid (5 wt%) had the lowest average effective heat of combustion (av-EHC) and the highest total smoke production (TSP), indicating that the addition of BP-Pid could improve the fire-inhibition activity in the gaseous phase. The above data revealed that the descending order of condensed phase flame retardancy efficiency was EP/BP-PPDA~EP/BP-DETA > EP/BP-Pid. Hence, the condensed-phase flame retardancy of aliphatic amine and aromatic amine modified BP in the epoxy matrix was dominant, while the cyclcamine modified BP mainly played the role of gas-phase flame retardancy.

## 2. Experimental Section

### 2.1. Materials

Red phosphorus (99.999%), diethylenetriamine (DETA), p-Phenylenediamine (PPDA), 4-amino-2,2,6,6-tetramethylpiperidine (Pid), thionyl chloride (SOCl_2_) and 4,4′-diaminodiphenyl sulfone (DDS) was purchased from Aladdin reagent corporation (Shanghai, China). Dimethyl formamide (DMF, 99.8%, extra dry, with molecular sieves) was purchased from Energy Chemical (Shanghai, China). Epoxy resin (EP, E-44) was supplied by Shandong Lutai Deyuan Epoxy Technology Co., Ltd. (Feicheng City, China).

### 2.2. Preparation of BP

In this work, the method of preparing BP by high-energy ball milling was referred to in the literature [48]. Red phosphorus (2 g) was sealed in a stainless vessel (100 mL) and ball-milled under Ar atmosphere for 12 h on a high energy mechanical milling (HEMM) machine. The inside pressure was kept at 1.6~1.8 MPa and the rotating frequency was set as 40 Hz. To avoid chemical degradation, the as-prepared BP was taken out and sealed in sample bottles directly in a vacuum glove box filled with Ar.

### 2.3. Preparation of NH_2_-Funtionalized of BP (BP-NH_2_)

Initially, BP was oxidized for a certain time and then uniformly dispersed in DMF (0.5 mg/mL, 50 mL) by ultrasonication. Secondly, 2.0 mL SOCl_2_ and 5.0 mL DETA were added into the BP suspension and the mixture was stirred at 80 °C for 12 h. After cooling, the suspension was centrifuged at 10,000 rpm for 30 min to obtain the residue. Then the residue was washed by DMF three times to remove the raw material and was dried in a vacuum oven at 50 °C for 48 h. The dried sample was sealed in a clean glass sample bottle before tests, named BP-DETA. All of the above reaction processes had been carried out in an inert atmosphere. BP-PPDA and BP-Pid were prepared with a similar process.

### 2.4. Preparation of EP/BP-NH_2_ Nanocomposites

Preparation process of EP nanocomposites with the addition of 1 wt% BP-NH_2_: 0.4 g of BP-NH_2_ was dispersed in 20 mL of acetone with the assistance of ultrasonication for 30 min. Following this, 31.2 g of EP was added into the above mixture with mechanical stirring at the corresponding temperature (e.g., T_DETA_ ≈ 150 °C) for 1 h. Subsequently, the whole system was placed in a vacuum to remove the acetone with the temperature of 60 °C for 2 h. After that, 8.4 g of DDS was melted and poured into above blends by a rapid stirring for 5 min. Finally, the resin sample was cured at 160 °C/1 h, 180 °C/2 h and 200 °C/1 h, respectively. After the curing process had finished, the EP/BP-NH_2_ (1 wt%) sample was permitted to cool to room temperature. A similar procedure was also used for pure EP, EP/BP-NH_2_ (3 wt%) and EP/BP-NH_2_ (5 wt%).

### 2.5. Characterizations

Laser Raman was performed by Super LabRam II system, Dilor, 532 nm He-Ne laser beams. The powder X-ray diffraction (XRD, D8-Advance instrument, Bruker AXS Co., Berlin, Gremany) was operated with Cu Kα radiation at a scan rate (20) of 5 °min^−1^ with an accelerating voltage of 40 kV. Fourier transform infrared (FTIR) spectra were performed by using a Nicolet IS50 spectrometer (Nicolet Instrument Co., USA). The samples were mixed with KBr powder and pressed into tablets before characterization. Thermo Gravimetric Analysis (TGA) was carried on DTG-60H (Shimadzu, Japan) from room temperature to 800 °C in N_2_/air at a heating rate of 10 °C·min^−1^. Scanning electron microscope (SEM) images were taken out on JSM-6700F field emission scanning electron microscope under the acceleration voltage of 3 kV. Transmission electron microscopy (TEM, Talos F200X) was performed to study the morphology of BP and BP-NH_2_. The chemical composition of materials was investigated by X-ray photoelectron spectroscopy (XPS) using an ESCALab250 electron spectrometer (Thermo Scientific Corporation) with monochromatic 150 W Al Kα radiation. Differential scanning calorimetry (DSC, Netzsch 200 F3) was used to test the exothermic peak with a heating rate of 5 °C min^−1^ under the protection of argon. Dynamic mechanical analysis (DMA) was employed to test the dynamic mechanical properties of EP/BP-NH_2_, all the samples were heated from room temperature to 250 °C with a linear heating rate of 5 °C min^−1^, and the frequency was 1 Hz for the tensile configuration. A cone calorimeter test (Fire Testing Technology Ltd., East Grinstead, UK) was performed to study the fire performance of EP composites according to the standard of ASTM E1354/ISO 5660 under the external heat flux of 50 kW/m^2^ [49]. The specimen dimensions were 100 × 100 × 3.2 mm^3^. The vertical burning test (UL-94) of the EP composites were performed by Horizontal and vertical burning instrument (CZF-4, Nanjing Shangyuan Analysis Instrument Co. Ltd., Nanjing, China) according to ASTM D3801 with the specimen dimensions of 125 × 12.5 × 3.2 mm^3^ [50]. In this test, the burning grade of a material was classified as V 0, V 1, V 2 or no rating (NR), depending on its behavior (dripping and burning time). The LOI tests were carried out in accordance with ASTM D2863 by oxygen index instrument (HC-2C, Nanjing Shangyuan Analysis Instrument Co. Ltd., Nanjing, China) [50]. The specimen dimensions were 125 × 6.5 × 3.2 mm^3^.

## 3. Results and Discussion

### 3.1. Characterizations of BP-NH_2_ Structure and Morphology

To unveil the structure of BP-NH_2_ compared to the BP nanosheets, a series of characterizations were performed. The FTIR was a common measurement to characterize the molecular structure due to the infrared vibrations of functional groups. Figure 2a showed the FTIR spectra of BP, BP-DETA, BP-PPDA and BP-Pid. Compared with the several characteristic peaks of BP nanosheets, two apparent peaks at 977 cm^−1^ and 905 cm^−1^ of BP-NH_2_ nanosheets were associated with the P–N–C and P–N characteristic absorption, respectively [51,52,53]. The bending vibration peaks of the –CH_2_-group were 1457 cm^−1^ and 720 cm^−1^, while the peak at 1381 cm^−1^ was ascribed to the –CH_3_ group. The peaks at 1255 cm^−1^ and 1184 cm^−1^ correspond to the stretching vibration of C–N of aromatic amine and aliphatic amine, respectively. The above IR peaks indicated that DETA, PPDA and Pid were chemically grafted to BP nanosheets via a P-N bond.

The structural integrity of the BP and BP-NH_2_ was further confirmed by XRD and Raman spectra, as shown in Figure 2b,c. The crystal structure of BP was an orthogonal crystal system, its XRD several representative diffraction peaks were 16.7°, 26.3°, 34.9°, 52.5° and 56.8° corresponding to the (020), (021), (111), (112) and (151) crystal planes, respectively, which were typical layered planes of BP. After surface amination, these peaks appear on the XRD of BP-DETA and BP-PPDA and were weaker and wider than that of BP, while the XRD peaks of BP-Pid (26.3°) existed in a left shift. Although the allotrope of P had the same valence state, the physical arrangement order of P element was different. In the Raman characteristic peak region of the P-P bond between 300 and 500 cm^−1^, it exhibited three representative vibrational modes of BP nanosheets, which were assigned to the peaks of A_g_^1^ at 357.2 cm^−1^, B_2g_ at 431.2 cm^−1^ and A_g_^2^ at 458.3 cm^−1^. During the process of surface amination, the vibration modes of BP did not change. Raman peaks of A_g_^1^, B_2g_ and A_g_^2^ were shown in Figure 2c. It was obvious that three peaks of the BP-NH_2_ nanosheets had a slightly blue shift (about 5–8 cm^−1^) compared to those of BP, due to the decreased thickness of the BP nanosheets [29]. XPS was evaluated to further probe the chemical valence bond and the extent of surface amination of BP. Figure 2d showed the survey spectra of BP and BP-NH_2_. Compared to the spectrum of BP, the elements content of N in the spectra of BP-NH_2_ increased. The N contents of BP-DETA, BP-PPDA and BP-Pid were 14.8%, 6.1% and 8.2% (Appendix A), respectively. Figure 2e presented the high-resolution P 2p XPS spectrum of BP nanosheets, which was deconvoluted into two peaks at 129.7 and 130.6 eV corresponding to P 2p_3/2_ and P 2p_1/2_ of P-P bonds, respectively. Figure 2f–h presented the high-resolution P 2p XPS spectra of BP–NH_2_. The two distinct peaks centered at 133.0 and 134.0 eV were consistent with P–N and P–O bonds, respectively [44,54,55], revealing the BP was inevitably oxidized during the experiment process and can be functionalized by amino compounds. The thermal stability of BP and BP-NH_2_ were studied under nitrogen by TGA. As shown in Figure 2i, BP-NH_2_ went through one-step decomposition, similar to BP. The onset thermal decomposition temperature and the temperature of maximum decomposition rate, were close to each other for BP and BP-NH_2_, as well. The surface amination of BP did not change the thermal stability of BP apparently. However, the residue amounts of BP-NH_2_ were much higher, at 700 °C. Especially, BP-DETA had 8% residue.

The microstructure of BP and BP-NH_2_ were characterized by TEM and SEM. As shown in the TEM image (Figure 3a), the exfoliated BP had a thinner lamellar structure with a size of several micrometers. Figure 3b showed the selected area electron diffraction (SAED) pattern image of the same exfoliated BP. The SAED pattern recorded on this sample depicted the good crystalline of the BP, and the high-resolution TEM images of BP (Appendix A) certified its layered or 2D structure. The lattice fringes of 0.33 nm corresponded to the (021) crystal plane of BP, which was in accordance with the results in the XRD analysis [56]. The SEM image of BP-DETA (Figure 3c) showed that the BP still well-maintained layered structure after the modification of amination. The energy dispersive spectrometer (EDS) was employed to characterize the element distribution in the BP-NH_2_ nanosheets. Figure 3d–f showed the BP-DETA nanosheets elemental mapping images, and it could be seen that C and N elements were uniformly distributed on the BP surface. The BP-PPDA and BP-Pid nanosheets elemental mapping images were shown in Appendix A. The successful formation of BP-NH_2_ nanosheets were further verified by elemental mapping images of phosphorus (P), nitrogen (N), and carbon (C).

### 3.2. The Amino Reactivity of BP-NH_2_ with Epoxide Group and the Dispersibility in EP

There was no doubt that the dispersion of nanofillers in polymer resins has a crucial influence on the mechanical properties and flame retardancy of polymer nanocomposites [57,58]. As a typical 2D inorganic nanofiller, the direct incorporation of BP into polymer will lead to agglomeration, which seriously restricts its application. Therefore, it is necessary to improve the dispersibility of BP in polymer resins. The NH_2_ group on BP-NH_2_ was designed to react with the epoxide group of epoxy resin, aiming to form chemical bond between BP and epoxy resin. DSC was used to study the reaction process of three amino groups of BP-NH_2_ and epoxy groups of EP. As shown in Figure 4a, the temperatures of exothermic peaks of BP-NH_2_/E44 were 150 °C, 180 °C and 220 °C, respectively, suggesting that the order of the activity of amino groups was: aliphatic amine > aromatic amine > cyclamine. During the curing process, the chemical bonding restricted the agglomerating behavior of BP, resulting in good dispersion of BP in EP. This could be analyzed by the fracture surface morphology. The SEM images of freeze-fractured surface for pure EP, EP/BP (5 wt%), and EP/BP-NH_2_ (5 wt%) nanocomposites were shown in Figure 4b–f. As shown in Figure 4b, the fractured surface of pure EP was extremely smooth and had a no-crinkled morphology. However, for the EP/BP (5 wt%) and EP/BP-NH_2_ (5 wt%) nanocomposites (Figure 4c–f), it could be seen that the fracture surfaces were rough and had small curling layers, which was caused by the ductile fracture of the EP caused by the addition of BP nanosheets. Obviously, BP without modification re-aggregated severely in EP resins with a poor dispersion, while BP-NH_2_ had a better dispersion in the EP nanocomposites. This result was also confirmed by the distribution of the P element EDS in the insert of Figure 4c–f.

### 3.3. Thermal Stability and Thermal Dynamic Mechanical Properties of EP Composites

The thermal and thermo-oxidative stability of pure EP and EP/BP-NH_2_ were studied by TGA under N_2_ and air respectively. The detailed data are shown in Table 1. In Figure 5a, EP/BP-NH_2_ nanocomposites had similar thermal degradation behavior to pure EP, which was a one-step weight loss process. Compared to pure EP, the onset degradation temperature (T_5%_) of the EP/BP-NH_2_ nanocomposites decreased. This was mainly due to the phosphoric acid compounds produced by BP-NH_2_, which reduced the activation energy of the EP matrix pyrolysis reaction. This promoted the dehydration, as well as the carbonization, of the matrix. T_max_ of EP/BP-NH_2_ shifted to around 370 °C. However, their char residues increased significantly at 800 °C, which was higher than that of pure EP. The residue amount of EP/BP-PPDA (5 wt%) were 25.2%, close to EP/BP-DETA (5 wt%), and slightly higher than that of EP/Pid (5 wt%). As can be seen from the dotted line area in Figure 5a, the carbonization of EP/BP-NH_2_ had a different stability at around 400 °C. That of EP/BP-PPDA was most stable; next was that of EP/BP-DETA. In Figure 5b, thermal-oxidative behavior of pure EP showed a two-stage curve under air. T_5%_ of pure EP was at 387 °C. T_max1_ and T_max2_ of pure EP were at 414 and 589 °C, respectively. There were almost no residues remaining at 800 °C. EP/BP-NH_2_ showed similar two-stage curves. Due to the catalyzing effect, T_d5%_ and T_max1_ of EP/BP-NH_2_ were lower than that of pure EP as well. With regard to the second DTG curves, T_max2_ of EP/BP-NH_2_ was higher than that of pure EP. In the thermal weightlessness stage of 400~800 °C, it showed that the residues of pure EP were not as stable as those of EP/BP-NH_2_ which were formed. At 800 °C, the residual weight of EP/BP-NH_2_ was higher than that of pure EP. That of EP/BP-PPDA was up to 6%. The results showed that the surface amination treatment of BP enhanced the charring ability of the EP matrix.

To study the influence of BP-NH_2_ on the thermomechanical properties of EP nanocomposites, the DMA was employed to test the storage modulus and tan δ of pure EP, EP/BP-NH_2_ composites [59]. The storage modulus and tan δ of pure EP and EP/BP-NH_2_ nanocomposites as a function of temperature were shown in Figure 5c,d. The storage modulus of pure EP at room temperature was 2912 MPa. The storage modulus of EP/BP-NH_2_ nanocomposite were slightly increased, mainly due to the enhancing effect of BP nanosheets. Compared with BP-DETA, BP-PPDA and BP-Pid were equipped with a ring structure and had a strong stiffness, which meant that EP had a higher storage modulus. The peak value of tan δ could be ascribed to glass transition temperature (T_g_) of the EP/BP-NH_2_ nanocomposites (Figure 5d). Along with the addition of BP-NH_2_ into an EP matrix, all the tan δ peaks were slightly moved to a lower temperature. The decrease in T_g_ was understood by the fact that the BP nanosheet increased the distance of polymer chains and decreased the crosslinking density of the matrix, leading to an increasing in the free volume.

### 3.4. Flame Retardancy of EP Composites

LOI was employed to evaluate the least oxygen concentration required for the combustion of the polymer, which could be used to quantitatively evaluate the fire performance, whereas the UL-94 test could more intuitively reflect the true combustion process of the material [60,61]. The corresponding results were listed in Table 2. The LOI value of pure EP was only 19.8% and its UL 94 test exhibited NR. The LOI values for EP/BP-NH_2_ were all notably enhanced with the incorporation of BP-NH_2_ content from 1 to 5 wt%. Compared with EP/BP-DETA and EP/BP-Pid, the augment of the LOI value of EP/BP-PPDA was more notably the same amount. The LOI value of EP/BP-PPDA (1 wt%) reached 27.9%, while those of BP-DETA (1 wt%) and EP/BP-Pid (1 wt%) were 26.1% and 26.2%, respectively. At 5 wt% loading, the LOI of the three EP/BP-NH_2_ were above 30%, and as high as 32.3%. Pure EP burnt out and was ranked as NR in the UL 94 test. When the addition of BP-NH_2_ in the EP matrix reached 5 wt%, the EP/BP-NH_2_ nanocomposite could successfully pass the UL 94 V 0 rating. Notably, EP/BP-Pid was no rating at 3 wt% loading, whereas the other two systems passed the V 1 rating. Figure 6 showed the digital photos of EP/BP-NH_2_ at different times during the UL 94 combustion test and no melt-drips were generated during the test. After leaving the 2nd ignition, EP/BP-NH_2_ (5 wt%) nanocomposite self-extinguished at 3.6 s, 1.8 s and 4.8 s, respectively. The impact of BP-Pid on self-extinguishment was not as good as those of BP-DETA and BP-PPDA.

Cone calorimeter tests, in accordance with the standard method ISO 5660, were adopted to further evaluate the burning behavior of EP in a realistic fire environment. The curves of the HRR, the THR, the TSP and weight of the EPs were shown in Figure 7. Table 3 lists the correlation characteristic parameters, such as the time to ignition (TTI), the time to the PHRR (t_PHRR_), the peak heat release rate (PHRR), the fire growth rate (FIGRA) and av-EHC. Pure EP was ignited at 40 s and burnt dramatically with high PHRR values of 1120 kW/m^2^. After incorporating BP-NH_2_ into EP, PHRR of EP/BP-NH_2_ nanocomposites decreased significantly. The PHRR of EP/BP-DETA was 605 kW/m^2^, decreasing by 46%. EP/BP/PPDA showed a reduction of 40%. EP/BP-Pid got a higher PHRR of 991 kW/m^2^. The difference was related to the catalytic effect of BP-NH_2_ on charring ability. Similar to the PHRR, THR had a similar trend. Pure EP was as high as 97.3 MJ/m^2^. By adding BP-NH_2_ at a content of 5 wt%, the THR of EP/BP-DETA (5 wt%), EP/BP-PPDA (5 wt%) and EP/BP-Pid (5 wt%) decreased directly to 51.6 MJ/m^2^, 47.0 MJ/m^2^ and 56.0 MJ/m^2^, respectively. Especially for EP/BP-PPDA (5 wt%), its THR was reduced by 51.70%. The above data revealed that the descending order of condensed-phase flame retardancy efficiency was EP/BP-PPDA > EP/BP-DETA > EP/BP-Pid.

The FIGRA was also employed to evaluate the fire hazard of the nanocomposites. The calculation of FIGRA was expressed as follows:


(1)
$$\mathrm{FIGRA}=\frac{PHRR}{t_{PHRR}}$$


Based on Table 3 and Equation (1), the FIGRA of pure EP was 13.3 kW m^−2^ s^−1^, whereas the FIGRA of EP/BP-DETA (5 wt%), EP/BP-PPDA (5 wt%) and EP/BP-Pid (5 wt%) nanocomposites were 5.2 kW m^−2^ s^−1^, 5.9 kW m^−2^ s^−1^, and 8.9 kW m^−2^ s^−1^, respectively. Particularly, the FIGRA of EP/BP-DETA (5 wt%) nanocomposites decreased by 60.9% in contrast to that of pure EP, which manifested that the fire growth rate was more effectively reduced by the addition of BP-DETA in the EP polymer. EHC was the heat of combustion, which would be expected in a fire where incomplete combustion takes place [52]. As presented in Table 3, it was found that the av-EHC values of EP/BP-NH_2_ nanocomposites decreased partly, as compared to that of pure EP. EP/BP-Pid (5 wt%) had the lowest av-EHC value of 12.4 kJ/kg. Generally, a low EHC value indicates noncomplete combustion, caused by the flame inhibition effect in the gas phase. The addition of the three BP-NH_2_ introduced a flame inhibition effect in the gas phase and they followed an effective order of BP-Pid > BP-PPDA ≈ BP-DETA.

The smoke release is one key factor in the cause of human death in a fire [62]. The addition of BP-NH_2_ reduced the smoke release of the EP matrix as well. EP/BP-PPDA had the lowest TSP of 20.2 m^2^/m^2^. The TSP values of EP/BP-DETA and EP/BP-Pid were 23 and 24.2 m^2^/m^2^, respectively. Compared with EP, the addition of BP-NH_2_ reduced the smoke release by around 48%, mainly benefiting the formation of char residue during the combustion. Three char residues, obtained from the combustion process, could provide an important perspective to investigate a reasonable flame-retardant mechanism. Figure 8 showed the digital photos of the external residues (a–d) from the top view and side view for EP and EP/BP-NH_2_ nanocomposites, as well as corresponding SEM images (e–h). Pure EP had a low charring ability, with a 9.2% char residue, which was loose and cracked (Figure 8a). Compared to pure EP, the carbon residue amount and quality of the EP/BP-NH_2_ nanocomposite were significantly increased. In Table 3, the char residue of EP/BP-DETA and EP/BP-PPDA were more than 30% residues. EP/BP-Pid had an 18.7% char residue. The good dispersion of BP nanosheets in EP was much conductive to the enhancement of the charring reaction. For BP-Pid, the tetramentylniperidine is one hindered amine, which might suppress the charring reaction of the radicals; hence, BP-Pid showed less char residues compared with BP-PPDA and BP-DETA. The char residues of EP/BP-NH_2_ were all intumescent shapes as shown in Figure 8b–d. The heights for EP/BP-Pid was 7.0 cm, slightly higher than that in other systems. The SEM images of the interior char layer showed that the interior char layer of EP containing BP-DETA and BP-PPDA nanocomposites were compact with few pores, whereas that of EP/BP-Pid was porous in Figure 8f–h. The graphitization degree of the char residue was studied by a Raman spectrum. A higher carbonization degree was more favorable to form an effective barrier to protect the internal materials from decomposing. Figure 9a–d showed the Raman spectra of EP and EP/BP-NH_2_ nanocomposites. It could be observed that two representative peaks of carbon materials were at 1370 cm^−1^and 1587 cm^−1^, defined as D peak and G peak, respectively. In prior studies, the ratio for D peak to G peak (I_D_/I_G_) was employed to assess the graphitization degree. A smaller I_D_/I_G_ represented a higher graphitization degree. The I_D_/I_G_ for pure EP was 3.49 (Figure 9a). The incorporation of BP-NH_2_ catalyzed the charring process to produce char residues with a high graphitization degree. In this respect, the EP/BP-Pid presented an I_D_/I_G_ of 2.91 (Figure 9d), whereas the EP/ BP-DETA and EP/BP-PPDA exhibited a lower I_D_/I_G_ (2.33 and 2.20) in Figure 9b,c. The above morphology and structure analysis indicated that the quality of the char residue for EP/BP-Pid was not as good as those of the other two systems. The consequences of this were consistent with the results of PHRR.

Moreover, XPS analysis was employed to study the element composition and chemical bonds of the char residues. Figure 10a exhibited the XPS spectra of pure EP, EP/BP-DETA, EP/BP-PPDA and EP/BP-Pid; the elements of C, N and O existed on the surface of all the samples. During the combustion of EP/BP-NH_2_, P elements were partly solidified to form phosphorus hybridizing char residues, which were confirmed by the peaks above 200 eV in Figure 10a. To show the bonding form, the high-resolution P 2p XPS spectrum for EP/BP-NH_2_ nanocomposites were collected in Figure 10b–d. The P 2p peaks for EP/BP-NH_2_ could be separated into three peaks at 133.3 eV, 134.3 eV and 135.4 eV, which were assigned to P-N, P-O and P_2_O_5_ bonds, respectively, indicating the formation of P_x_O_y_ and P-N bonds in the char residues. For EP/BP-DETA, the peak intensity of P_2_O_5_ was close to those of P-O and P-N bonds. It followed that the structures of the surface amination of BP impacted on the charring process of the EP matrix. On the whole, BP-DETA accumulated an EP matrix to form more char residues with a dense structure, a high graphitization and an intumescent shape, next to BP-PPDA, resulting in the best insulation effects on heat and gaseous compounds. Therefore, PHRR and av-EHC trends were intelligible for the three systems.

### 3.5. Flame Retardant Mechanism

Based on the above results and analysis for combustion behavior in EP/BP-NH_2_ nanocomposites, a flame retardation mechanism was described in Figure 11. The same as other phosphorus-containing flame retardant, three kinds of BP-NH_2_ affected the combustion behavior of EP both in gas and condensed phases. During the decomposition of BP-NH_2_, free radicals such as PO· released into the gas phase, scavenging H· and OH· radicals and showing a flame inhibition effect. On the other hand, phosphoric acid derivatives formed by P_x_O_y_ and water from the dehydration of polymer chains promoted the carbonization process (Figure 8), acting as an isolator of gaseous compounds and heat exchange (the PHRR and THR were decreased as shown in Figure 7a,b. In the meantime, combustible gases were reduced owing to the formation of more char residue in the gas phase. The difference for the three kinds of BP-NH_2_ was in the efficiency in the two phases. BP-DETA and BP-PPDA promoted the charring quality and quantity of EP matrix, while BP-Pid showed an enhanced flame inhibition effect. The results indicated that the modification structure can influence the BP flame behavior in the polymer matrix.

## 4. Conclusions

In this work, three amine-containing compounds (aliphatic amine, aromatic amine and cyclamine) were chosen to functionalize BP via the nucleophilic substitution reaction, and its structure and composition was confirmed by FT-IR, XRD, Raman, XPS, and TEM. The reactivity of amino groups of BP-NH_2_ and epoxy groups were different. The results showed that the amino group of BP was modified by aliphatic amine and had the highest reactivity with the epoxy group, and the descending order of reactivity between amino and epoxy groups was EP-BP-DETA > EP-BP-PPDA > EP-BP-Pid. The surface amino functionalization method was dramatically solved by the aggregation phenomenon and enhanced the dispersibility of the BP in EP matrix, thereby improving the flame retardancy efficiency. The HRR, THR and TSP values of EP/BP-NH_2_ nanocomposites are significantly decreased and its LOI value was dramatically improved. The V 0 rating was achieved at 5wt% loading of BP-NH_2_ in EP. The relationship between the amino structures and the flame retardancy efficiency of condensed-gas phases of BP in the EP matrix were compared. Among the above three typical amino-contain compounds, the condensed-phase flame retardancy of aliphatic amine and aromatic amine modified BP in the epoxy matrix was dominant, while the cyclcamine modified BP mainly played the role of gas-phase flame retardancy.

## Figures and Tables

**Figure 1 polymers-13-03635-f001:**
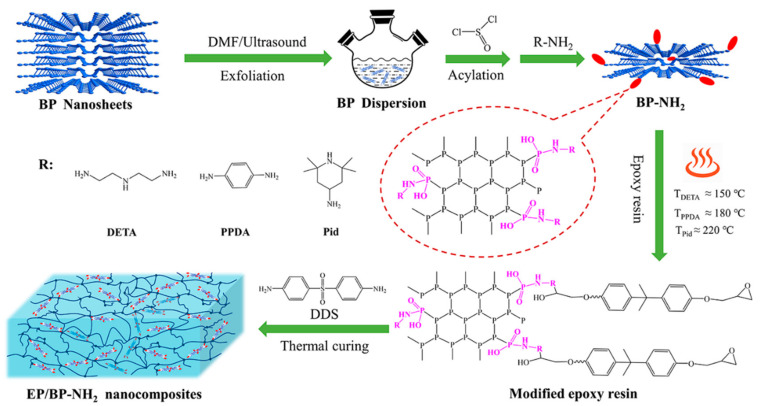
The preparation process of BP-NH_2_ nanofiller and EP/BP-NH_2_ nanocomposites.

**Figure 2 polymers-13-03635-f002:**
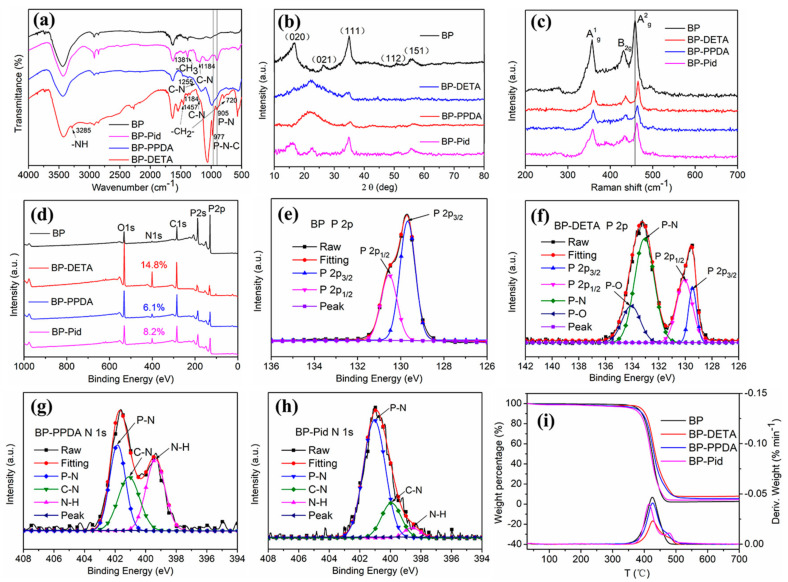
Characterization of BP and BP-NH_2_: (**a**) FT-IR spectra; (**b**) XRD patterns; (**c**) Raman spectra; (**d**) XPS survey spectra and high-resolution P 2p XPS spectra of (**e**) BP nanosheets, (**f**) BP-DETA, (**g**) BP-PPDA, and (**h**) BP-Pid; (**i**) TGA and DTG curves.

**Figure 3 polymers-13-03635-f003:**
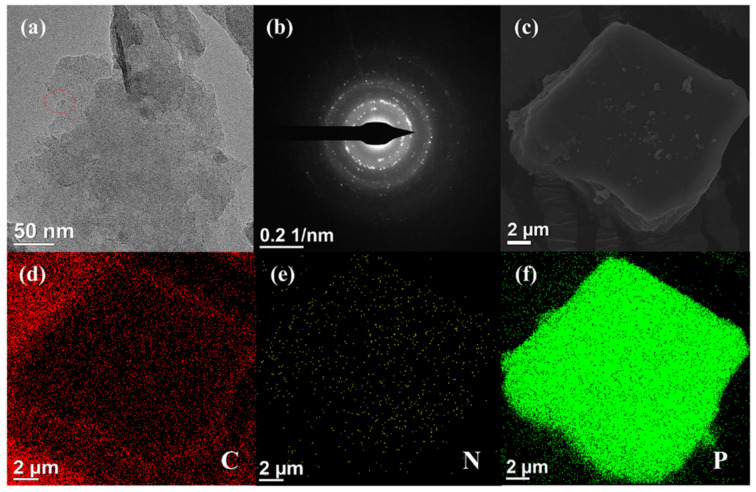
(**a**) TEM image and (**b**) SAED pattern of the BP nanosheets; (**c**) SEM image of BP-DETA nanosheets and the corresponding elemental mapping images of (**d**) carbon (C), (**e**) nitrogen (N), (**f**) phosphorus (P).

**Figure 4 polymers-13-03635-f004:**
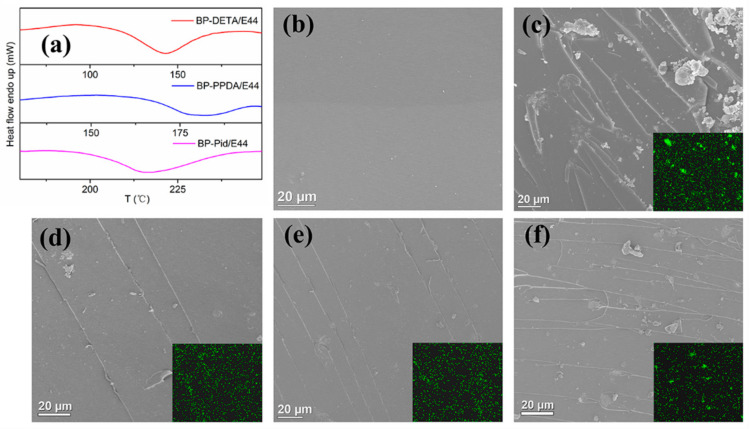
(**a**) DSC thermograms of BP-NH_2_/E44; SEM images of freeze-fractured surface for (**b**) pure EP, (**c**) EP/BP (5 wt%), (**d**) EP/BP-DETA (5 wt%), (**e**) EP/BP-PPDA (5 wt%), (**f**) EP/BP-Pid (5 wt%) nanocomposites and the corresponding P element distribution diagram.

**Figure 5 polymers-13-03635-f005:**
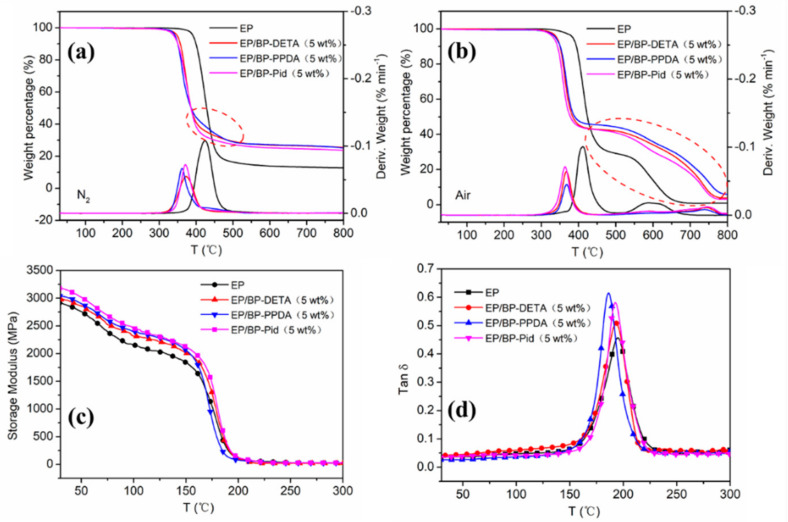
TGA and DTG curves of the pure EP and EP/BP-NH_2_ nanocomposites under N_2_ (**a**) and under air (**b**); DMA results of the pure EP and EP/BP-NH_2_ nanocomposites: storage modulus E’ (**c**,**d**) tan δ curves.

**Figure 6 polymers-13-03635-f006:**
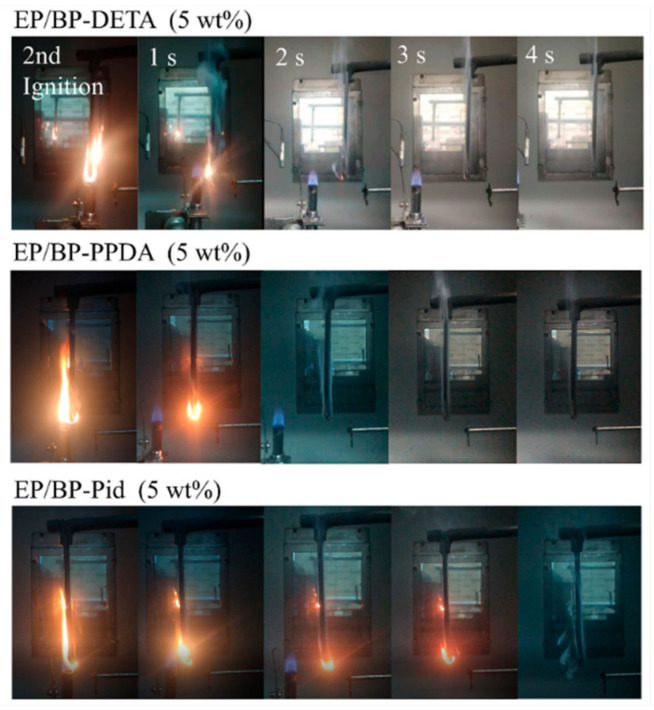
The digital photos of EP/BP-NH_2_ at different times during the UL 94 combustion test.

**Figure 7 polymers-13-03635-f007:**
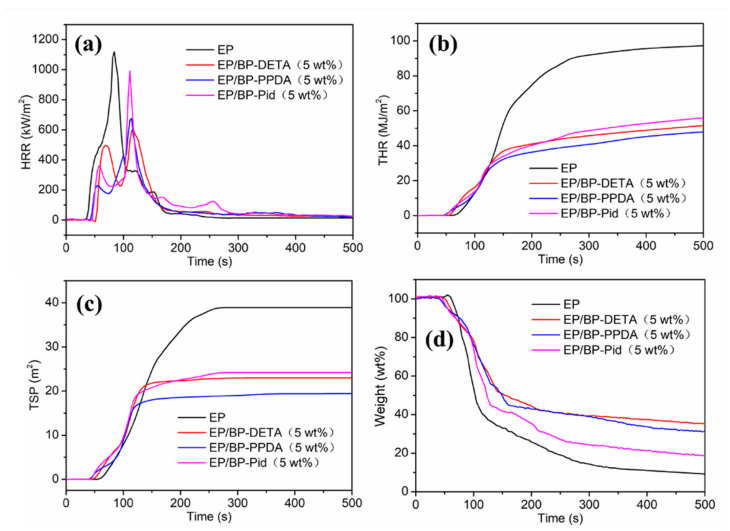
(**a**) HRR, (**b**) THR, (**c**) TSP and (**d**) Weight versus time curves of the pure EP and EP/BP-NH_2_ nanocomposites from cone calorimeter tests.

**Figure 8 polymers-13-03635-f008:**
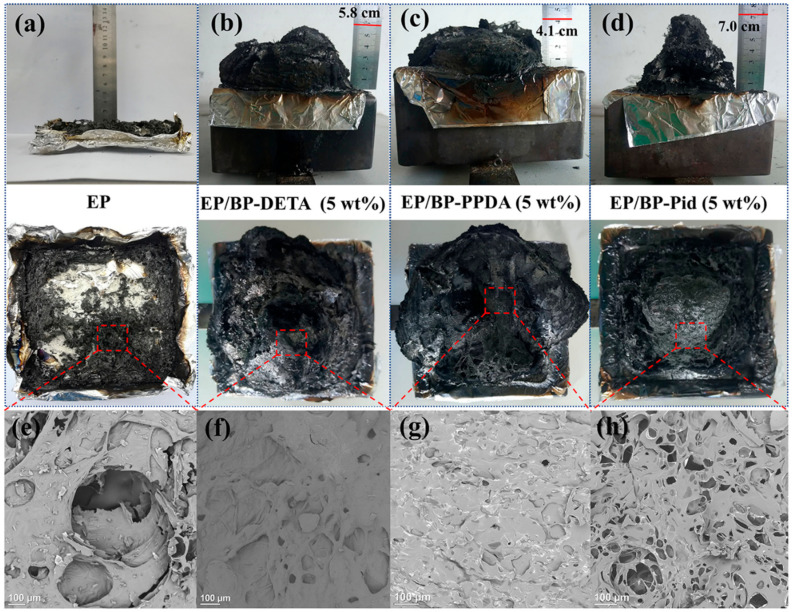
Digital photos of the external residues (**a**–**d**) from a top view and a side view for pure EP and EP/BP-NH_2_ nanocomposites; SEM images of interior (**e**–**h**) char residues for pure EP and EP/BP-NH_2_ nanocomposites.

**Figure 9 polymers-13-03635-f009:**
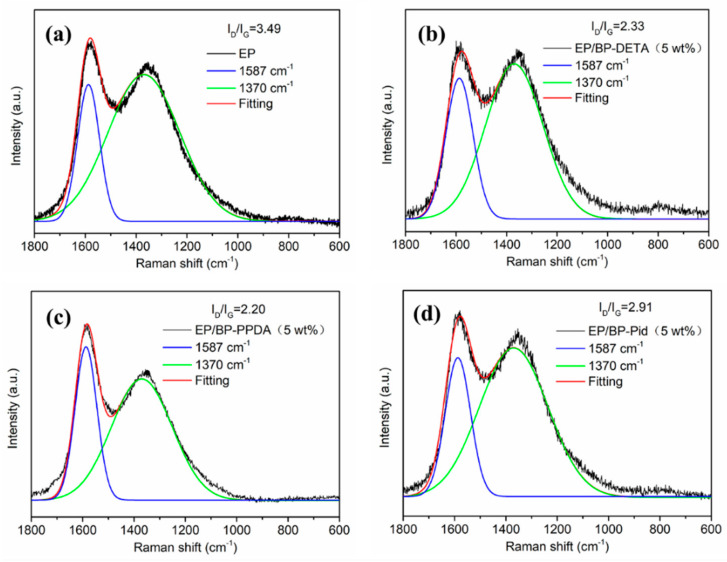
Raman spectra of the char residues of (**a**) pure EP, (**b**) EP/BP-DETA (5 wt%), (**c**) EP/BP-PPDA (5 wt%) and (**d**) EP/BP-Pid (5 wt%).

**Figure 10 polymers-13-03635-f010:**
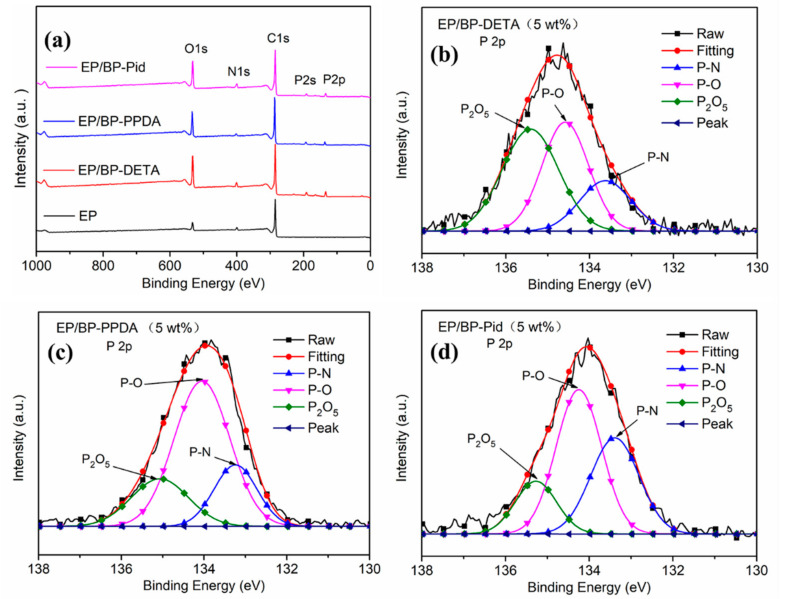
(**a**) XPS survey spectra of the residual char for pure EP and EP/BP-NH_2_ nanocomposites after cone tests; high-resolution P2p XPS spectra of (**b**) EP/BP-DETA (5 wt%), (**c**) EP/BP-PPDA (5 wt%), and (**d**) EP/BP-Pid (5 wt%).

**Figure 11 polymers-13-03635-f011:**
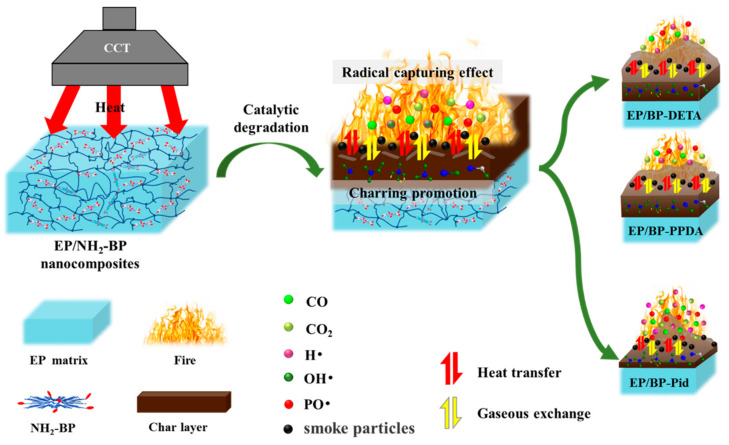
Flame-retardant mechanism of BP-NH_2_ in EP.

**Table 1 polymers-13-03635-t001:** TGA data of pure EP, EP/BP-NH_2_ under N_2_ and air.

Sample	T_5%_ ^a^ (°C)		T_max1_ ^b^ (°C)		T_max2_ ^c^ (°C)	Residue (800 °C, %)	
	N_2_	Air	N_2_	Air	Air	N_2_	Air
	391	385	425	411	588	12.8	0.9
	349	342	374	365	739	25.5	3.8
	343	339	363	366	739	25.2	6.0
	339	333	371	363	747	23.7	3.1

Note: The 5% in the sample names meant the weight percentage. ^a^ T_5%_ meant the temperature when the weight loss was 5%. ^b^ T_max1_ meant the first peak T at maximum decomposition rate. ^c^ T_max2_ meant the second peak T at the maximum decomposition rate.

**Table 2 polymers-13-03635-t002:** LOI and UL-94 vertical burning test data of pure EP and its nanocomposites.

Sample	LOI (%)	UL 94		
		$\bar{t_{1}}$	$\bar{t_{2}}\;(s)$	Rating
EP	19.8	>30	>60	NR
EP/BP-DETA (1 wt%)	26.1	>30	>60	NR
EP/BP-DETA (3 wt%)	28.2	7.3	11.8	V 1
EP/BP-DETA (5 wt%)	30.1	1.8	3.6	V 0
EP/BP-PPDA (1 wt%)	27.9	>30	>60	NR
EP/BP-PPDA (3 wt%)	30.1	2.0	10.6	V 1
EP/BP-PPDA (5 wt%)	32.3	1.6	1.8	V 0
EP/BP-Pid (1 wt%)	26.2	>30	>60	NR
EP/BP-Pid (3 wt%)	27.8	15.0	42.3	NR
EP/BP-Pid (5 wt%)	31.9	3.0	4.8	V 0

$\bar{t_{1}}$ and $\bar{t_{2}}$ were the remaining flame times after the first and second ignition respectively, and the ignition time was 10 s each time.

**Table 3 polymers-13-03635-t003:** The data from the cone calorimeter test of pure EP and EP/BP-NH_2_ nanocomposites at a heat flux of 50 kW/m^2^.

Sample	TTI ^a^ (s)	T_PHRR_ ^b^ (s)	PHRR ^c^ (kW/m^2^)	FIGRA (kW/m^2^/s)	THR ^d^ (MJ/m^2^)	av-EHC ^e^ (kJ/kg)	TSP ^f^ (m^2^)	Residue (wt %)
EP	40	84	1120	13.3	97.3	15.2	38.9	9.2
EP/BP-DETA (5 wt%)	40	117	605	5.2	51.6	13.5	23.0	35.1 (↑73.8%)
EP/BP-PPDA (5 wt%)	37	114	674	5.9	47.0	13.2	20.2	31.3 (↑70.6%)
EP/BP-Pid (5 wt%)	38	111	991	8.9	56.0	12.4	24.2	18.7 (↑50.8%)

^a^ TTI was time to ignition. ^b^ T_PHRR_ time to the PHRR. ^c^ PHRR was peak of heat release rate. ^d^ THR was total heat release. ^e^ av-EHC meant average effective heat of combustion (between 40 and 300 s). ^f^ TSP meant the total smoke production.

## Data Availability

The data presented in this study are available on request from the corresponding author.

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
