# Peer review of "Surface Functionalization of Black Phosphorus via Amine Compounds and Its Impacts on the Flame Retardancy and Thermal Decomposition Behaviors of Epoxy Resin"

_polymers, 2021, doi:10.3390/polym13213635_

Round 1

Reviewer 1 Report

In the submitted paper three different amine-based compounds have been used to functionalize black phosphorus which was added to epoxy resin to improve its flame retardancy.

The synthesis of the materials has been described and a comprehensive characterization has been performed by several analytical techniques. Flammability has been also investigated and a possible flame-retardant mechanism has been proposed.

I suggest just a few minor changes before the publication.

- Figures 2 are difficult to read; please, increase the size.

- Check and revise the sentences at lines 195-196, 197-199, 340-341, and 356-357.

- The distribution maps of P, reported in Figures 4, are not clearly visible; try to change the contrast.

- Line 246: to avoid misinterpretation, “mechanical” should be replaced with “dynamic mechanical”.

- Figure 5: I would suggest using two separate figures for TGA-DTG and DMA results. If no, the label at lines 270-272 needs to be revised.

Reviewer 2 Report

The manuscript concerns functionalization of BP and its impact on the epoxy resin. The manuscript is interesting, good prepared. Experiments and  analyzes well planned and executed. Insightful interpretation of the obtained results.

The Introduction, the third paragraph 49-64; in the first sentence the authors cite a large amount of references on how to improve BP stability. I miss publication references in the further part, when discussing specific solutions.

2.5. Characterizations; Please add the relevant references to the standards used.

2.5. Characterizations; The dimensions of the samples used in the tests are a little unclear. I understand that most of the tests required 100x100x3.2 samples, UL-94 test required 125.5x12.5x3.2 specimen and LOI test 125.5x6.5x3.2. Last sentence: "The dimensions for each test were 125.5×6.5×3.2mm3." can be confusing, I suggest to improve the description.

Please also standardize the temperature notation throughout the manuscript, the degree symbol is used in different formatting.
